

# *Acptp2,3* participates in the regulation of spore production, stress response, and pigments synthesis in *Aspergillus cirstatus*

Lei Shao[1,2], Zuoyi Liu[3,4,5] and Yumei Tan[3,4]

[1] College of Pharmacy, Guiyang Healthcare Vocational University, Guiyang, Guizhou, China
[2] Guiyang Healthcare Vocational University, Guizhou Provincial Engineering Research Center of Medical Resourceful Healthcare Products, Guiyang, Guizhou, China
[3] Guizhou Key Laboratory of Agricultural Biotechnology, Guiyang, Guizhou, China
[4] Institute of Biotechnology, Guizhou Academy of Agricultural Sciences, Guiyang, Guizhou, China
[5] Innovative Institute for Plant Health, Zhongkai University of Agriculture and Engineering, Guiyang, China

Corresponding author
Yumei Tan, 540764039@qq.com

## ABSTRACT

**Background**. *Aspergillus cristatus* was a filamentous fungus that produced sexual spores under hypotonic stress and asexual spores under hypertonic stress. It could be useful for understanding filamentous fungi's sporulation mechanism. Previously, we conducted functional studies on *Achog1*, which regulated the hyperosmotic glycerol signaling (HOG) pathway and found that SI65_02513 was significantly downregulated in the transcriptomics data of Δ*Achog1* knockout strain. This gene was located at multiple locations in the HOG pathway, indicating that it might play an important role in the HOG pathway of *A. cristatus*. Furthermore, the function of this gene had not been identified in Aspergillus fungi, necessitating further investigation. This gene's conserved domain study revealed that it has the same protein tyrosine phosphatases (PTPs) functional domain as *Saccharomyces cerevisiae*, hence SI65_02513 was named *Acptp2,3*.

**Methods**. The function of this gene was mostly validated using gene knockout and gene complementation approaches. Knockout strains exhibited sexual and asexual development, as well as pigments synthesis. Morphological observations of the knockout strain were carried out under several stress conditions (osmotic stress, oxidative stress, Congo Red, and sodium dodecyl sulfate (SDS). Real-time fluorescence polymerase chain reaction (PCR) identified the expression of genes involved in sporulation, stress response, and pigments synthesis.

**Results**. The deletion of *Acptp2,3* reduced sexual and asexual spore production by 4.4 and 4.6 times, demonstrating that *Acptp2,3* positively regulated the sporulation of *A. cristatus*. The sensitivity tests to osmotic stress revealed that Δ*Acptp2,3* strains did not respond to sorbitol-induced osmotic stress. However, Δ*Acptp2.3* strains grew considerably slower than the wild type in high concentration sucrose medium. The Δ*Acptp2,3* strains grew slower than the wild type on media containing hydrogen peroxide, Congo red, and SDS. These findings showed that *Acptp2,3* favorably controlled osmotic stress, oxidative stress, and cell wall-damaging chemical stress in *A. cristatus*. Deleting *Acptp2,3* resulted in a deeper colony color, demonstrating that *Apctp2,3* regulated pigment

synthesis in *A. cistatus*. The expression levels of numerous stress-and pigments-related genes matched the phenotypic data.

**Conclusion**. According to our findings, *Acptp2,3* played an important role in the regulation of sporulation, stress response, and pigments synthesis in *A. cristatus*. This was the first study on the function of PTPs in Aspergillus fungi.

## INTRODUCTION

*Aspergillus cristatus* was a naturally occurring probiotic fungus that showed a "flowering" process. Pure sexual and asexual spores could be prepared under laboratory circumstances, indicating osmotic stress primarily regulated the sporulation of *A. cristatus* (*Liu & Qin, 1991*). However, the relationship between osmotic stress and sporulation had not been completely investigated. It was well understood that the hyperosmotic glycerol mitogenic kinase (HOG) signaling pathway was one of the primary ways that eukaryotic cells responded to osmotic stress (*Gustin et al., 1998*; *O'Rourke, Herskowitz & OShea, 2002*). The *Achog1* gene was found to be homologous to *hog1* of *S. cerevisiae* in the genome database of *A. cristatus* and its function was examined. The transcriptome sequencing of the Δ*Achog1* strains revealed a highly down-regulated gene, SI65_02513, which was annotated on *ptp2,3* of the pheromone-MAPK pathway, cell wall stress-MAPK pathway, HOG-MAPK pathway, and hunger-MAPK pathway in the HOG pathway. According to domain analysis, the gene had a "protein tyrosine phosphatase (PTP) fungal protein" superfamily domain, which comprises tyrosine phosphatase 1 (PTP1) and tyrosine phosphatase 2 (PTP2) in *S. cerevisiae*, as well as fungal proteins PTP1, PTP2, and PTP3 in *Saccharomyces pombe*. PTPs were a catalyst that could eliminate phosphotyrosine peptides and hence alter phosphotyrosine concentrations in signal transduction pathways. Various studies were conducted on PTPs in different organisms, results showed that PTPs played a crucial role in how cells responded to physiological and pathological changes in their environment (*Kanner, 2020*). In addition, tyrosine phosphatase is directly associated to human diseases. For example, cancer or improper cell death has been linked to inappropriate tyrosine phosphatase phosphorylation in cells (*Hunter, 2009*). For plants, PTPs could contribute in abscisic acid (ABA), exogenous calcium, darkness, and $H_2O_2$ stress, which leads to stomatal closure in plants (*MacRobbie, 2002*; *Kanner, 2020*). PTP was first examined in *S. cerevisiae*, where Ptp2 and Ptp3 were found to dephosphorylate proteins involved in the HOG pathway (*Mattison et al., 1999*). So yet, little was known about the role of PPTs in Aspergillus fungus.

Given the importance of PTPs in humans, plants, and *S. cerevisiae*, it was assumed that PTPs would also play a crucial role in filamentous fungi, and that this gene was involved in the HOG pathway of *A. cristatus*. To test the effects of *Acptp2,3* on *A. cristatus*, homologous recombination was employed to create the gene knockout vector and complement vector. The *Acptp2,3* gene knockout strain and complementation strain

were created *via Agrobacterium tumefaciens*-mediated transformation (ATMT-mediated transformation). The effect of *Acptp2,3* on growth and development was studied by comparing the morphological changes between Δ*Acptp2,3* knockout strains and the wild type of *A. cristatus*.

## MATERIALS & METHODS

### Experimental materials

The pDHt-sk-*hyg* and pDht-sknt plasmids used in this study were stored in our laboratory. The wild type (WT) strain of *A. cristatus* (CGMCC 7.193) was isolated from Fuzhuan brick tea made by the Yiyang Tea Factory in Yiyang, China. To induce sexual and asexual development, the strains grew on MYA solid medium with low osmotic stress at 28 °C, and on on MYA solid medium with high osmotic stress at 37 °C (20 g of malt extract, 5 g of yeast extract, 30 g of sucrose, 170 g of sodium chloride, and 1000 mL of water, *Shao et al., 2022*), respectively. Using IM ($K_2HPO_4$ 2.05 g; $KH_2PO_4$ 1.45 g; NaCl 0.15 g; $MgSO_4 \bullet 7H_2O$ 0.5 g; $CaCl2 \bullet 6H_2O$ 0.1 g; $FeSO_4 \bullet 7H_2O$ 0.0025 g; $(NH_4)_2SO_4$ 0.5 g; glucose 2.0 g, 0.5%(W/V)glycerol) as the induction medium for transformation. The morphological observation medium consisted of MYA medium supplemented with various quantities of sucrose, sorbitol, hydrogen peroxide, Congo red, and SDS chemicals. The strains were photographed with a camera (Canon EOS 7D Mark II; Canon, Tokyo, Japan).

### Experimental methods

#### Analysis of Actp2,3 sequence

The amino acid sequences of Ptp2,3 protein was downloaded from the NCBI database, including *A. cristatus* (ODM21669.1), *Aspergillus glaucus* (XP_022401591.1), *Aspergillus ruber* (XP_040636285.1), *Aspergillus melleus* (XP_045945710.1), *Didymosphaeria variabile* (XP_056068816.1), *Purpureocillium takamizusanense* (XP_047848236.1), *Candida albicans* (XP_719371.1) and *S. cerevisiae*. MEGA 6.06 software was used to perform phylogenetic analysis on the *Acptp2,3* proteins. ClustalW (maximum likelihood (ML) analysis was performed using RAxML-HPC BlackBox tool CIPRES in web portal and the default GTRGAMMA + I model) was used to align multiple sequences with the default values. A phylogenetic tree was created with maximum likelihood and a bootstrap value of 1,000 (*Miller, Pfeiffer & Schwartz, 2010*).

### Screening and identification of the Δ*Acptp2,3* and Δ*Acptp2,3-C* strain

*Acptp2,3*'s whole CDS was deleted using homologous recombination techniques. The *Acptp2,3* deletion cassette featuring *hph* (a resistance gene of hygromycin) as the selective marker was created by fusing the 5′-untranslated region (5′-UTR) and 3′-untranslated region (3′-UTR) of the *Acptp2,3* gene. To construct the final knockout vector, up-*Acptp2,3*-pDHt/sk-*hyg*-*Acptp2,3*-down, the 5′-*BamH* I-*Xho* I UTR and 3′-*Spe* I-*Xba* I UTR were amplified using appropriate primer pairs from genomic DNA of the WT strain and cloned into the cloning sites of pDHt/sk-*hyg* plasmid (a schematic diagram of vector construction as shown in Fig. S1). To complement the *Acptp2,3* mutant, the *Acptp2,3* gene with its own promoter was amplified from genomic DNA using primers (Table S1) and

inserted between *Hind* III and *Kpn* I of the pDHt/sknt plasmids. The plasmid sequences were confirmed by polymerase chain reaction (PCR), restriction enzyme digestion, and sequencing. The transformation was carried out exactly as described earlier (*Tan, 2008*). *Agrobacterium tumefaciens* strain LBA4404, comprising a previously produced vector, was cultivated in a liquid minimum medium at 28 °C for 48 h, with 50 ug/mL kanamycin. A 100 uL aliquot of the *A. tumefaciens* fluid solution was combined with an equal volume of a conidial suspension from the WT strain or Δ*Acptp2,3* strain and cultivated at 28 °C for approximately 9 h. The mixture was plated on an IM plate and cultured at 28 °C for 48 h. MYA medium with 300 ug/mLAmpicillin, 50 ug/mL Hygromycin B, or 200 ug/mL Geneticin (G418) was plated on a coculture plate at 28 °C until transformants developed. The transformed strains were validated using both PCR and real-time quantitative PCR (RT-qPCR), the primers design principles required for the validation of Δ*Acptp2,3* and Δ*Acptp2,3-C* strains were shown in Figs. S2 and S3.

## Morphological observation

Various strains were inoculated on the matching medium, colony morphology was observed, colony diameter was measured, sexual and asexual spore production statistics, and various sensitivity tests in Δ*Acptp2,3* and Δ*Acptp2,3-C* were performed.

## Statistical analysis

Statistical analysis of gene expression, colony diameter, ascospore and conidia number: Set up three biological replicates, three measurements were recorded and the data were analyzed statistically. * $p < 0.05$: the lowest significance; ** $p < 0.01$: a moderate degree of significance; *** $p < 0.001$: the highest significance.

## Real time-PCR detection

Real time-PCR (RT-PCR) was carried out as previously described in *Shao et al. (2022)*. Total RNA was extracted at the tested time point. Then, 2 mg of RNA was utilized to synthesize cDNA using a RevertAid First Strand cDNA Synthesis Kit (catalogue#K1622; Thermo Fisher Scientific, Waltham, MA, USA). RT-qPCR was performed using a CFX96 Real-Time PCR Detection System (Bio-Rad Laboratories, Hercules, CA, USA) in a total volume of 10 uL, which consisted of 5 uL of SsoFast EvaGreen SuperMix (catalogue # 172-5,201; Bio-Rad Laboratories), 1 uL of each primer (10 pmol/mL and 1 uL of template). *GAPDH* was selected as a candidate reference gene. The primers employed in the RT-qPCR experiments were designed using the Primer 3 online program and the resulting RT-qPCR products were tested *via* agarose gel electrophoresis. Each primer pair was tested with serial dilutions of cDNA to determine the linear range of the RT-qPCR assays. Three biological replicates were analysed (*Shao et al., 2022*). All the RT-qPCR primers used were listed in Table S1.

## RESULTS AND ANALYSIS

### Screening and identification of the Δ*Acptp2,3* and Δ*Acptp2,3-C* strain

Analysis of the gene sequence of *Actp2,3* showed that *ptp2,3* was highly conserved among Aspergillus specifications and *ptp2,3* homologs as a well supported group with *A. Melleus*

(Fig. S4). And it has a PTP-fungal domain spanning amino acids 512-777, which was identical to the conserved domain of the *ptp2,3* gene in *S.cerevisiae*. Subsequently, the Δ*Acptp2,3* knockout strain was constructed, and the enzyme digestion results showed that the size of the product was expected, indicating that the plasmid up-*Acptp2,3*-pDHt/sk-*hyg-Actp2,3*-down (Fig. S5A) and the complementation vector were successfully constructed (Fig. S52B). Δ*Acptp2,3* knockout and Δ*Acptp2,3-C* complementation strains were obtained through ATMT-mediated transformation (*Tan, 2008*). Specific primers (the primers used in this study are shown in Table S1) were used to validate Δ*Acptp2,3* and Δ*Acptp2,3-C* strains. The segments amplified from the WT and Δ*Acptp2,3* genomic DNA templates were 3,160 bp and 4,206 bp respectively (Fig. S6A). RT-PCR was used to further validate the Δ*Acptp2,3*, and the results showed that the expression of the *Acptp2,3* gene was not detected in the knockout strains (Fig. S6B). A total of 13 Δ*Acptp2,3-C* strains were confirmed by PCR (Fig. S6C). Further detection by RT-PCR revealed that the expression level of *Acptp2,3* was nearly identical to that of the WT strain, indicating that *Acptp2,3* was actually complemented in Δ*Acptp2,3* strains (Fig. S6D). The above results revealed that knockout strains might be utilized to investigate phenotypic differences.

### *Acptp2,3* promoted the production of ascospores and conidia in *A. cristatus*

Studies showed that the absence of *ptp2,3* leaded to the defection of sporulation in mutants, even the mutants almost lost the ability of sporulation (*Yang, 2013*). To investigate the impact of *Acptp2,3* on sexual and asexual sporulation, the conidial liquid ($1.0 \times 10^6$/mL) of wild strains WT, Δ*Acptp2,3*, and Δ*Acptp2,3-C* were inoculated on 1 M MYA solid medium and cultivated at 28 °C for 7 days. Phenotype and microstructure were examined. The results revealed that the wild-type strains and Δ*Acptp2,3* strains could continue on sexual development. There was no significant change in the shape of the cleistothecium on MYA medium with 1 M NaCl (Fig. 1A, columns 1–2). The cleistothecium was crushed and examined under a microscope. On the 14th day of culture, both Δ*Acptp2,3* and wild-type strains generated typical ascospores (Fig. 1B). However, the wild-type strain produced 4.4 times more ascospores than the Δ*Acptp2,3* strains (Fig. 1C). Similar findings were obtained in MYA medium with 1 M sorbitol and 1 M sucrose (Fig. 1A, columns 4–6).

The conidial liquid ($1.0 \times 10^6$/mL) of wild strains WT, Δ*Acptp2,3*, and Δ*Acptp2,3-C* were inoculated on 3 M MYA solid medium and cultivated at 37 °C for 7 days to investigate the effect of Δ*Acptp2,3* strains on asexual sporulation. The diameter of Δ*Acptp2,3* was smaller than that of the wild type (Fig. 2B), and the colony was yellow-green (Fig. 2A, first row). The conidial output was roughly 4.6 times lower than that of the wild type on the seventh day (Fig. 2C). Microscopic analysis revealed no significant variation in asexual sporulation structure between WT, Δ*Acptp2,3*, and Δ*Acptp2,3-C* (Fig. 2A, row 2).

### Deleting *Acptp2,3* gene effects the response of *A. cristatus* to high osmotic stress

As a key gene in the HOG pathway, one of the main functions of *ptp2,3* is to regulate the osmotic stress of *S. cerevisiae* (*Mattison & Ota, 2000*). WT, Δ*Acptp2,3*, and Δ*Acptp2,3-C* strains were grown in MYA medium with 2 M/3 M sorbitol/sucrose at 37 °C. Δ*Acptp2,3*

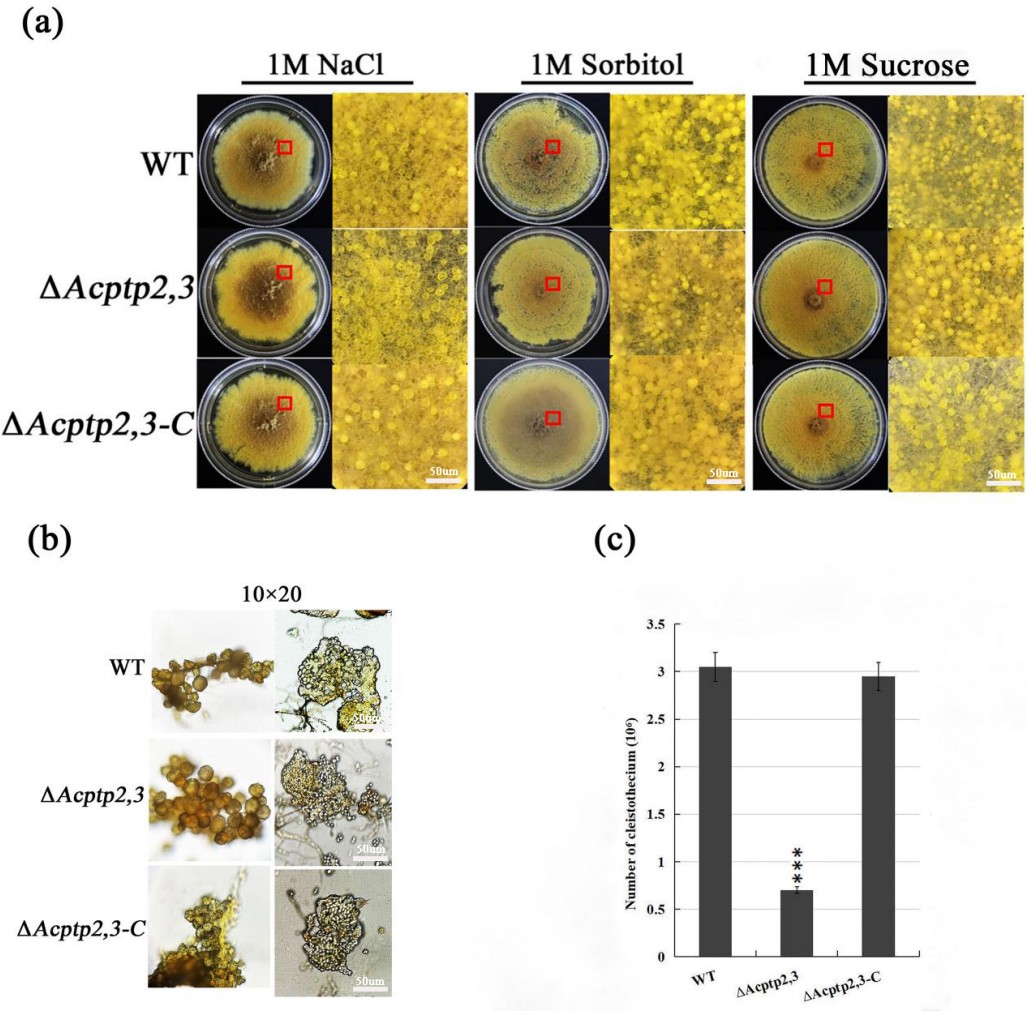

**Figure 1  Sexual sporulation of WT, ΔAcptp2,3, and ΔAcptp2,3-C.** (A) Microscopical and colony morphology of MYA in 1 M sodium chloride, sorbitol, and sucrose. (B) Cleistothecium and ascospores of WT, ΔAcptp2,3, and ΔAcptp2,3-C. (C) Statistics on ascospore production by WT, ΔAcptp2,3, and ΔAcptp2,3-C, ***$p = 0.000000001$.

grew similarly to the wild type, but at a slower rate on MYA medium with 2 and 3 M sorbitol (Figs. 3A–3B). The conidial number of WT, ΔAcptp2,3, and ΔAcptp2,3-C strains were statistically examined on a medium with 3 M sorbitol and 3 M sucrose. In the presence of sorbitol and sucrose, ΔAcptp2,3 produced considerably less conidia. Wild-type strain produced 3.14 and 2.2 times more conidia than ΔAcptp2,3 strains on MYA medium with 3 M sorbitol and sucrose (Fig. 3C). The results showed that ΔAcptp2,3 was more susceptible to high sucrose concentrations than sorbitol. Furthermore, Acptp2,3 might increase the conidial production of A. cristatus in sorbitol and sucrose-rich mediums.

### Role of *Acptp2,3* under oxidative stress

When conducting functional studies on the *Fgptp* gene in *Fusariurri graminearurr*, it was found that Δ*Fgptp* strains were not sensitive to oxidative stress (*Jiang, 2012*). We would

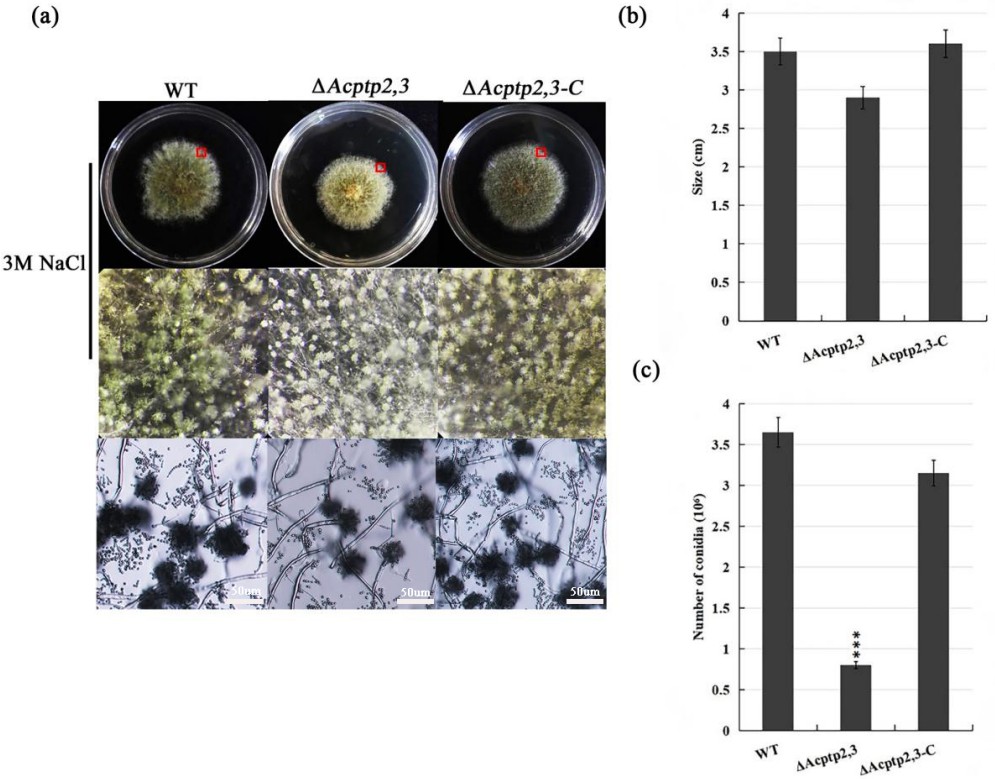

**Figure 2** **Asexual sporulation of WT, Δ*Acptp2,3*, and Δ*Acptp2,3-C*.** (A) Colony and microscopic morphology of WT, Δ*Acptp2,3*, and Δ*Acptp2,3-C* on MYA with 1 M sodium chloride, sorbitol, and sucrose. (B) Statistics of colony diameters for WT, Δ*Acptp2,3*, and Δ*Acptp2,3-C*. (C) Conidial number for WT, Δ*Acptp2,3* and Δ*Acptp2,3-C*, \*\*\**p* = 0.000000001.

like to know the effect of *Actp2,3* on the response of oxidative stress in *A. cristatus*. WT, Δ*Acptp2,3*, and Δ*Acptp2,3-C* conidial suspensions were inoculated on MYA medium with hydrogen peroxide ($H_2O_2$), whereas MYA medium without any agent was utilized as the control medium. Oxidative stress affects Δ*Acptp2,3* strains. The colony diameter of Δ*Acptp2,3* was substantially smaller than that of the wild type on medium with varied doses of $H_2O_2$ (10 mM, 30 mM, and 50 mM) (Figs. 4A–4B). The results showed that Δ*Acptp2,3* was much more susceptible to $H_2O_2$, indicating that this gene might modulate the response of oxidative stress in *A. cristatus*.

### *Acptp2,3* was involved in the response of Congo red and sodium dodecyl sulfate in *A. cristatus*

In *F. graminearurr*, the absence of *Fgptp2* slowed down the growth of Δ*Fgptp2* strain (*Jiang, 2012*). We wanted to know if there would be similar results in the absence of *Actp2,3*. WT, Δ*Acptp2,3*, and Δ*Acptp2,3-C* were inoculated on MYA medium (0.5 M NaCl) including agentia to detect the sensitivity of Δ*Acptp2,3* to Congo red and SDS. MYA (0.5M NaCl) with no additions was employed as the control medium. The results demonstrated that both Δ*Acptp2,3* and WT could grow on the medium containing Congo red. After 5

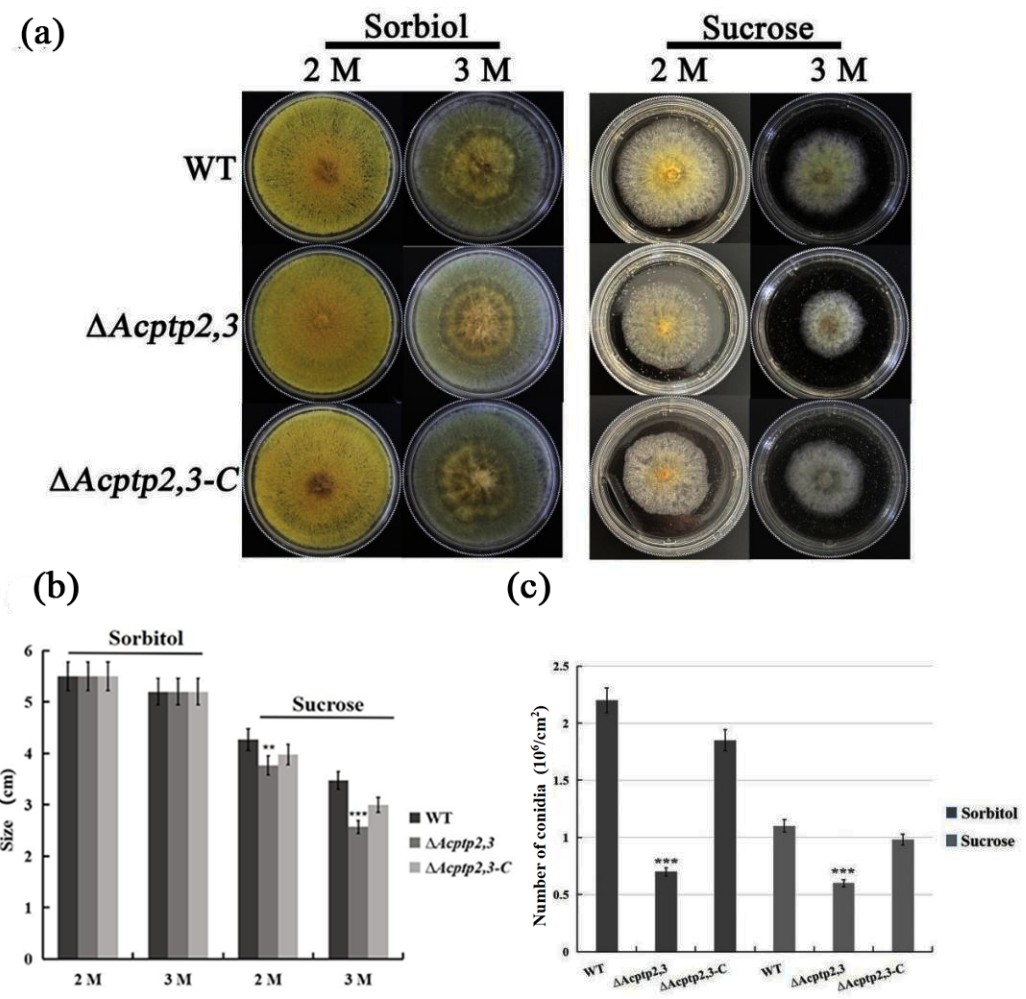

**Figure 3** **The effects of *Acptp2,3* on the osmotic stress response.** (A) Colony morphology of wild-type strains, Δ*Acptp2,3* and Δ*Acptp2,3-C* under various osmotic stresses. (B) Colony diameter statistics for wild-type strains Δ*Acptp2,3* and Δ*Acptp2,3-C* under various osmotic stress conditions. (C) Conidia production statistics for wild-type, Δ*Acptp2,3*, and Δ*Acptp2,3-C* strains under various osmotic stress conditions, **$p = 0.002569$, ***$p = 0.0000003$.

days of cultivation, the Δ*Acptp2,3* strains had wider colony diameters than the WT strains, indicating reduced sensitivity to Congo red (Figs. 5A and 5B). WT and Δ*Acptp2,3-C* strains struggled to grow in a medium with 0.01% SDS, whereas Δ*Acptp2,3* strains performed well (Figs. 5C and 5D). These findings revealed that *Acptp2,3* was involved in the response to cell wall-damaging chemicals such as Congo red and SDS.

## Deleting *Acptp2,3* caused colonies to deepen in color

WT, Δ*Acptp2,3*, and Δ*Acptp2,3-C* were inoculated on MYA solid medium to compare pigments. As illustrated in Fig. 6A, the WT colony's center was brown, with bright yellow margins. The Δ*Acptp2,3* colony had a brown center and yellow margins. The colony color of Δ*Acptp2,3-C* resembled that of the wild type (Fig. 6A). RT-qPCR was used to detect

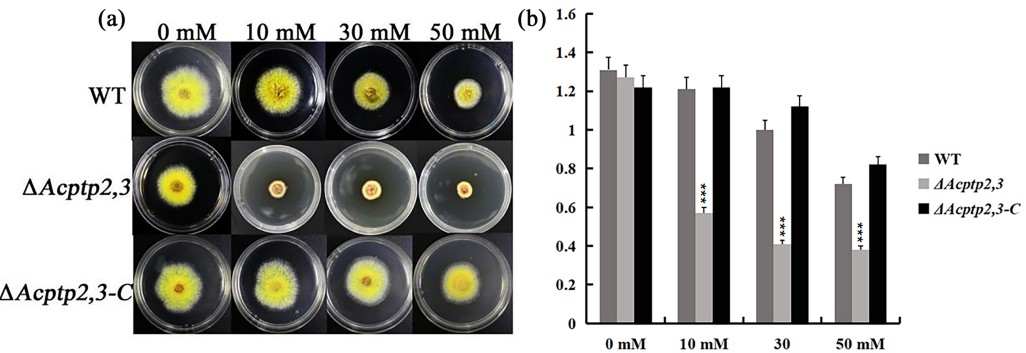

**Figure 4** **The effect of *Acptp2,3* on the oxidative stress response.** WT and Δ*Acptp2,3* were grown in MYA medium with varying doses of hydrogen peroxide at 28 °C for 3 days. (A) Different $H_2O_2$ concentrations affect the morphology of Δ*Acptp2,3* colonies. (B) Δ*Acptp2,3* colony diameters at varied $H_2O_2$ concentrations. 10 mM: \*\*\**p* = 0.0000007; 30 mM: \*\*\**p* = 0.0000001; 50 mM: \*\*\**p* = 0.000004.

the expression levels of four genes (SI65_08209, SI65_08202, SI65-08742 and SI65_08211) involved in pigments synthesis. The Δ*Acptp2,3* strain had considerably greater expression levels of four pigments synthesis genes compared to the wild-type and Δ*Acptp2,3-C* strains (Fig. 6B), demonstrating that *Acptp2,3* could decrease the pigments of *A. cristatus*.

### RT-qPCR analysis of related gene expression

The wild type strains grew faster than Δ*Acptp2,3* strains in high sucrose and $H_2O_2$ medium, but slower in Congo red and SDS media. We used real-time fluorescence PCR to detect the expression of associated genes. The expression levels of genes associated with osmotic stress and oxidative stress were found to be lower than those of the wild type, while genes associated with the Congo red response were notably higher (Fig. 7). These findings indicated that *Acptp2,3* adversely regulated the expression of genes associated with osmotic stress and oxidative stress while positively regulating the expression of genes associated with the Congo red response.

## DISCUSSION

The *Acptp2,3* gene of *A. cristatus* contained a "PTPs superfamily" domain. In *S. cerevisiae*, *ptp2* and *ptp3* dephosphorylated proteins involved in the HOG pathway (*Mattison et al., 1999*). In addition, *ptp2* and *ptp3* caused aberrant spore formation in *S. cerevisiae* (*Zhan & Guan, 1999*). In other fungi, a small number of studies suggested that PTP was critical for fungal spore formation. In *F. graminearurr*, deleting *Fgptp2* caused the number of spores fell dramatically, even the sporulation was nearly impossible in Δ*Fgptp2* (*Jiang, 2012*). After knocking out *BcptpA* in *B. cinerea*, the development rate of the knockout strain reduced and the ability to generate spores was nearly completely lost (*Yang, 2013*). In *Colletotrichum graminicola*, deleting *CgptpM1* resulted in delayed conidial germination and three times fewer conidia than that of wild type (*Wang, 2021*). It was suggested that tyrosine phosphatase was engaged in some signaling pathways during spore germination. During the process, PTPs altered the dynamic balance of kinase phosphorylation, regulating

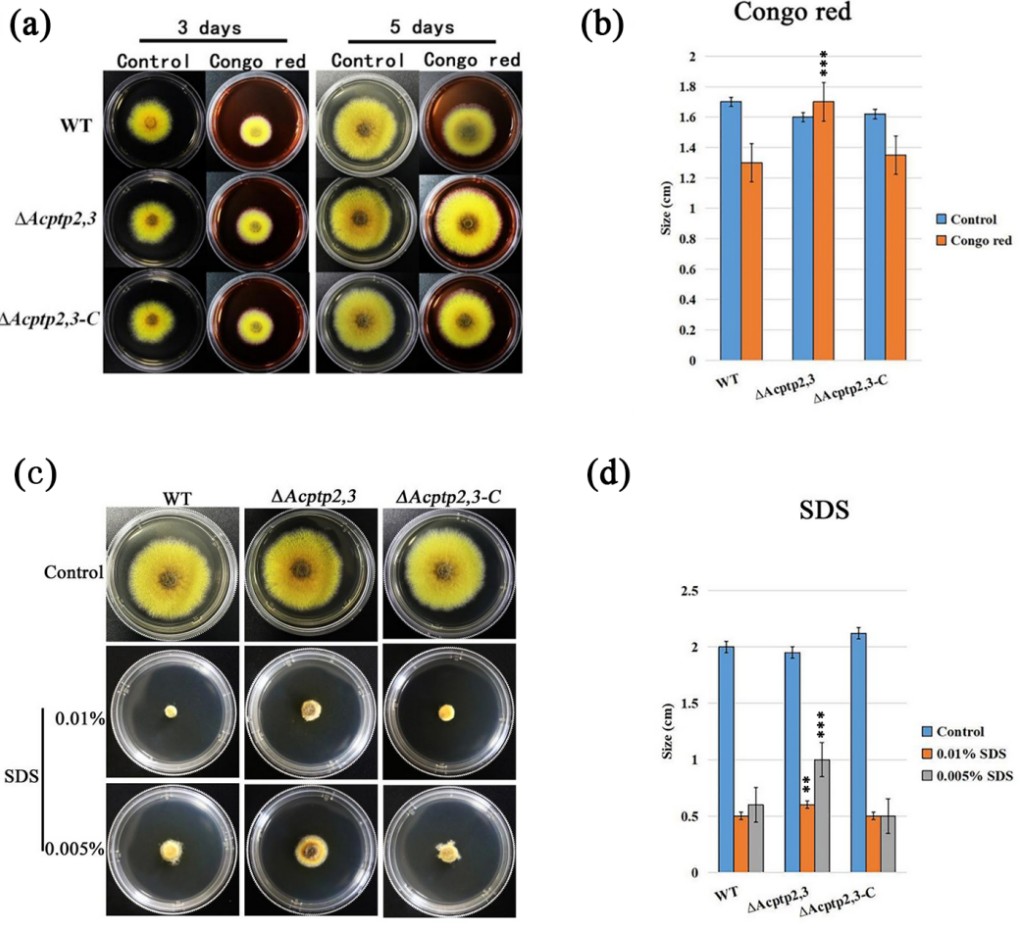

**Figure 5** *Acptp2,3* **affects the reaction of Congo red and sodium dodecyl sulfate.** Δ*Acptp2,3* colony morphology on Congo red-containing media. (B) Colony diameter statistics for Δ*Acptp2,3* in Congo red medium, ***$P = 0.00009$. (C) Δ*Acptp2,3* colony morphology on medium with varying sodium dodecyl sulfate concentrations. (D) Statistics of colony diameters Δ*Acptp2,3* on media with different concentrations of sodium dodecyl sulfate, **$p = 0.049$; ***$p = 0.00006$.

germination and sporulation. In our investigation, deleting *Acptp2,3* in *A. crsitatus* resulted in a significant reduction in sporulation.

Δ*Acptp2,3* strains produced 4.4 and 4.6 times fewer ascospores and conidia than that of wild type. The expression levels of gene related to sporulation showed that *Acptp2,3* reduced sporulation by downregulating the expression of SI65_05591, SI65_10255, and SI65_05589 in *A. cristatus*. In addition, prior research revealed that the *MAT* mating gene was the major gene influencing sexual sporulation, while the BrlA-AbaA-WetA central regulatory pathway was the primary route influencing asexual sporulation in *A. cristatus* (*Ge et al., 2016*). *Acptp2,3* was thought to regulate the *MAT* gene and the *BrlA-AbaA-WetA* pathway, affecting both sexual and asexual sporulation in *A. cristatus*.

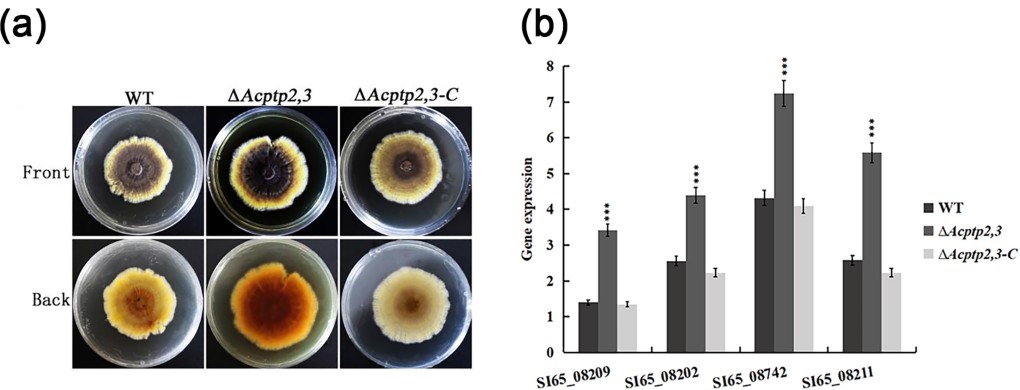

**Figure 6** **Pigmentation of the WT, ΔAcptp2,3, ΔAcptp2,3-C strains.** (A) Pigmentation of the WT, ΔAcptp2,3, ΔAcptp2,3-C strains. (C) Expression of genes related to pigment synthesis. SI65_08209: ***$p = 0.0003$; SI65_08202: ***$p = 0.00000004$; SI65_08742: ***$p = 0.000000005$; SI65_08211: ***$p = 0.0000000003$.

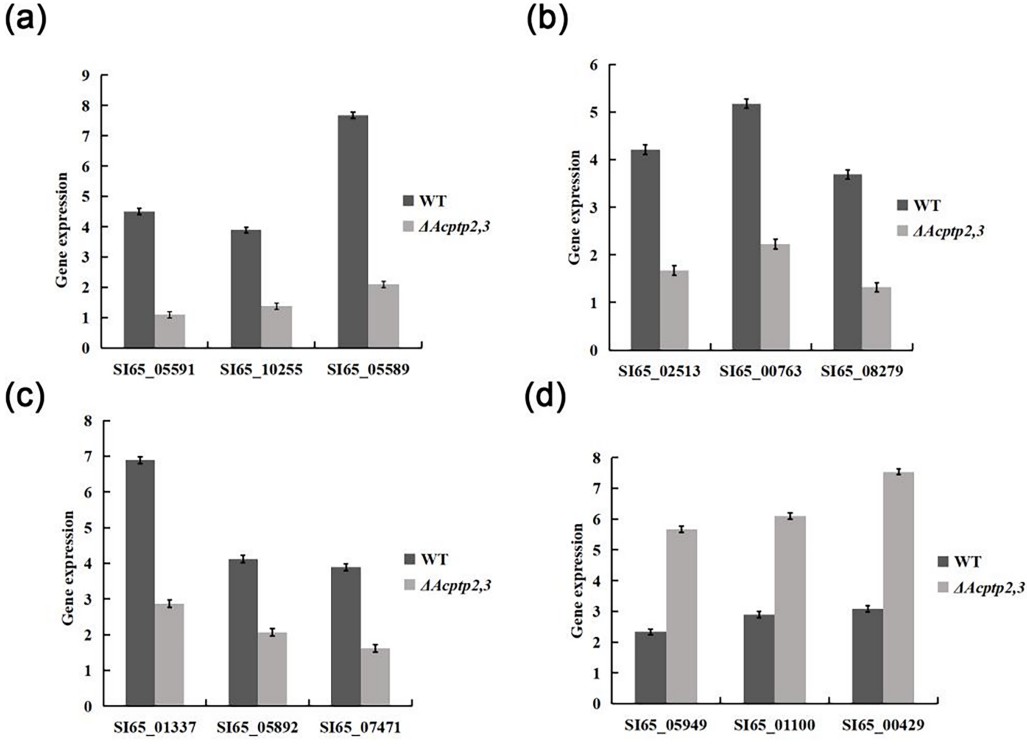

**Figure 7** **RT-qPCR analysis of gene expression.** (A) Expression of genes involved in sporulation. (B) Expression of genes involved in osmotic stress. (C) Expression of genes involved in oxidative stress. (D) The expression of genes involved in cell wall integrity.

In terms of environmental stress response, the growth of ΔBcptpA and ΔBcptpB was significantly slower than that of the wild type on PDA plates containing NaCl and KCl in *B. cinerea*, but they were not sensitive to the osmotic stress produced by sorbitol and

glucose. The growth of ∆*BcptpA* and ∆*BcptpB* on plates containing Congo red and caffeine was inhibited (*Yang, 2013*). In *C. graminicola*, ∆*CgptpM1* exhibited strong response to NaCl, sorbitol, and $H_2O_2$ (*Wang, 2021*). In order to investigate the effect of *Acptp2,3* on the response of *A. cristatus* to environmental stress, osmotic stress, oxidative stress, Congo red, and SDS sensitivity tests were conducted on ∆*Acptp2,3*. The results showed that ∆*Acptp2,3*, like ∆*BcptpA* and ∆*BcptpB*, did not respond to the osmotic stress produced by sorbitol, but was sensitive to the osmotic stress produced by NaCl and sucrose. Under oxidative stress, the growth of ∆*Acptp2,3* slowed down. Unlike the PTPs in *F. graminearurr* and *C. graminicola*, the ∆*Acptp2,3* grew faster than the wild-type on media containing Congo red and SDS. The TCHK (Two-component histidine kinase) signaling pathway was implicated in the regulation of osmotic stress in *B. cinerea* (*Yang, 2013*), but the HOG system was the major mechanism in response to osmotic stress in *A. cristatus*, with *Achog1* as essential regulatory genes. *Actp2,3* was speculated to be the target gene of *Achog1* (*Shao et al., 2022*). In this study, we used yeast two hybrid technology to confirm the interaction of the two genes. Therefore, it was speculated that *Actp2,3* could interact with *Achog1* to regulate the response of osmotic stress. There have been indications indicating the $Ca^+$ and MAPK signaling pathways are critical for the integrity of cell walls in fungi (*Xu, Staiger & Hamer, 1998*; *Rebollar & López-García, 2013*). We predicted that *Acptp2,3* would influence the response to cell wall-damaging chemicals by interacting with key genes in these two regulatory pathways in *A. cristatus*. In addition, based on the expression levels of genes related to stress response, it was also speculated that *Acptp2,3* might interact with genes such as SI65_02513, SI65_014337, and SI65_05949 to regulate the response of environmental stress. Our findings suggested that *A. cristatus* had a distinct osmotic stress pathway. PTPs had only a minor effect on strain pigments in *B. cinerea*; however, the deletion of *BcptpA* and *BcptpB* resulted in an increase in colors produced by knockout strains (*Yang, 2013*). In *A. cristatus*, the lack of *Actp2,3* resulted in an increase in pigment. The expression levels of four pigment related genes were detected, and the results showed that *Acptp2,3* could reduce the pigment of *A. cristatus* by downregulating the expression of SI65_08209, SI65_08202, SI65_08742 and SI65_08211 genes.

In this study, we first reported the function of *ptp2,3* in Aspergillus fungi. On the one hand, our findings contributed to the research of PTPs in fungi; on the other hand, they were the first to report the function of PTPs in Aspergillus, which may provide some reference significance for the study of PTPs in other Aspergillus.

## CONCLUSIONS

This work used gene knockout and complementation approaches to investigate the function of the *Acptp2,3* gene. The findings revealed that *Acptp2,3* stimulated the production of sexual and asexual spores, favorably regulated osmotic and oxidative stress, and negatively regulated the stress of cell wall damaging substances in *A. cristatus*. In addition, *Acptp2,3* reduced pigment synthesis in *A. cristatus*.

## ACKNOWLEDGEMENTS

We especially thank Chengshu Wang (Institute of Plant Physiology and Ecology, Shanghai Institute for Biology Science, Chinese Academy of Science) for providing plasmid pDHt/sk-*hyg*.

### Funding

This research was funded by the Guiyang Healthcare Vocational University Doctoral Fund (No. Guiyang Healthcare Vocational University K2024-8), the Central Government Guidance for Local Science and Technology Development Projects for Guizhou Province (No. [2023]027), the Youth Foundation of Guizhou Academy of Agricultural Sciences (No. [2023]05), and the Science and Technology Foundation of Guizhou Province (No. qiankehejichu-ZK [2022] general482). There was no additional external funding received for this study. The funders had no role in study design, data collection and analysis, decision to publish, or preparation of the manuscript.

### Grant Disclosures

The following grant information was disclosed by the authors:
The Guiyang Healthcare Vocational University Doctoral Fund: Guiyang Healthcare Vocational University K2024-8.
The Central Government Guidance for Local Science and Technology Development Projects for Guizhou Province: [2023]027.
The Youth Foundation of Guizhou Academy of Agricultural Sciences: [2023]05.
The Science and Technology Foundation of Guizhou Province: qiankehejichu-ZK [2022] general482.

### Competing Interests

The authors declare there are no competing interests.

### Author Contributions

- Lei Shao conceived and designed the experiments, performed the experiments, authored or reviewed drafts of the article, and approved the final draft.
- Zuoyi Liu conceived and designed the experiments, authored or reviewed drafts of the article, and approved the final draft.
- Yumei Tan performed the experiments, analyzed the data, prepared figures and/or tables, authored or reviewed drafts of the article, and approved the final draft.

### Data Availability

The data is available at NCBI: JXNT01000002.1.

## Supplemental Information

Supplemental information for this article can be found online at http://dx.doi.org/10.7717/peerj.17946#supplemental-information.

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
