# Peer review of "Acptp2,3 participates in the regulation of spore production, stress response, and pigments synthesis in Aspergillus cirstatus"

_PeerJ, doi:10.7717/peerj.17946_

## Round 0.1 · original submission · Major Revisions

Three experts assessed your manuscript and found merit in the manuscript content to be considered for publication in this journal. However, several concerns need to be addressed before providing a positive decision.

Among the points raised by the Reviewers, it is relevant to address the inclusion of more experiments, generating a control strain transformed with the empty vector to demonstrate the phenotypes are not associated with its structural components. The high level of self-citation and details about the qPCR assays are also aspects that I would expect to be solved in a revised version of the manuscript.

Reviewer 1 ·

Basic reporting

In general, the manuscript follows a dynamic that is easy to follow. However, there are some suggestions and fixes.

Throughout the manuscript there are some references that should be added.

Experimental design

Some of the methods (I will specify them later) need to be better detailed and referenced.

Although the approach to answer the research question is appropriate. I suggest a transformation of Aspergillus with the empty plasmid; This will help define that the changes obtained are due to the lack of the gene and not to the effects of the plasmid.

Validity of the findings

It seems that the data shown support the hypothesis of the participation of the gene under study for Aspergillus. But again, I consider that for the data to be completely valid, it is necessary to show the results of the mushroom transformation with the empty vector.

Additional comments

The name of the microorganism in the title does not coincide with that shown throughout the manuscript.

Define (HOG, PTPs, SDS, ABA)

I suggest you put keywords

In the first part of the introduction (lines 48-52) there are incomplete ideas and he combines them with each other. Restructure them.

Line 63: Similar in what?

74-74: Reference needed.

87: Fullmane of ATMT- transformation

99: ml --> mL and uL change to the apropiete symbol.

112, 258, 260: check the correct writing of all microorganisms

113: post more information about Clustal W (on which server?)

118: Reference needed.

119: hph definition

116: better describe this protocol

155-156: reference needed

165-178: Are you talking about bacteria or fungus?

208-214: Reference needed.

227: replace the word infected

244-247: reference needed.

Reviewer 2 ·

Basic reporting

The writing can be substantially improved. There are very confusing paragraphs, partly due to the use of language but also due to the inappropriate use of concepts. The concepts of Knockout and Knockdown are used interchangeably as synonyms when they are genetically different concepts. Another example is the pigmentation synthesis referred to in line 30. The pigmentation is not synthesized; what is synthesized are the pigments. This issue is repeated at several other points in the text.
In line 63 said that “The Achog1 gene was determined to be similar to hog1 of S. cerevisiae”, similar or homologs?
Line 64 “gene belonged to the PTP-fungal protein family,”. Genes do not belong to protein families.
Line 81 “Ptp2 and Ptp3 have been shown to dephosphorylate the HOG pathway”. Dephosphorylate the HOG pathway or proteins that participate in the HOG pathway?
Lines 86 -87. Knockout or Knockdown?
Line 112. S. cerevisia?
The sections: “Sequence analysis of gene Acptp2,3”;“Construction of AcPtp2,3 knockout and complementation vectors”; and “Identification of the Acptp2,3 deletion strain and complementation strain” could be addressed without problem in the material and methods section
According to the above, table 1 and Figures 1, 2, and 3 could be presented as supplementary material.
The discussion section could be restructured; it is long and only recounts the published results of the study of homologous proteins in other fungi. The possible regulatory mechanisms, as well as their control targets, are not discussed. The results section shows the work where the regulatory effect of the studied protein is determined; it is said that they are genes related to the processes controlled by Apctp2,3. But it needs to be said what genes they are and why they were chosen, which should be addressed in the discussion section.

Experimental design

The paper complies with the journal requirements in terms of aims and scope. However, there is a lack of information regarding qRT-PCR experiments. It is not reported how the quantitative PCR experiments were carried out, the data processing that was carried out, the internal controls that were used, etc.

Validity of the findings

The conclusions section is a summary. The paragraphs 301 to 304 contain the conclusions of the work. The rest can be part of the discussion

Additional comments

in general, the work needs to improve the quality of language and writing. There are also conceptual errors, in addition to other points to improve the structure of the article.

Reviewer 3 ·

Basic reporting

Results
Line 168 and 169.- mention that the Wt strain and ΔAcptp2,3 genomic DNA templates were 4206 bp and 3160 bp, respectively. However, Figure 3A shows that in the red boxes the Acptp2,3 mutants are obtained, and the amplification size observed is 4206 bp.
Figure 3.- In the expression levels indicate with which sample the data were normalized.
Figure 4 and 5 .- scale of measurement, observation objective, bright field, etc. (Microscopy characteristics).
Figure 6 to 9.- .- scale of measurement.
They have more than 40% of self-citation. Justify.

Experimental design

Include more details in a graph of the deletion system used.

Validity of the findings

Materials and methods.
Add the section on statistical analysis performed.

---

## Round 0.2 · accepted · Accept

The authors addressed all the Reviewers' concerns and the manuscript is now suitable for publication in this journal.